# scMGCN: A Multi-View Graph Convolutional Network for Cell Type Identification in scRNA-seq Data

**DOI:** 10.3390/ijms25042234

**Published:** 2024-02-13

**Authors:** Hongmin Sun, Haowen Qu, Kaifu Duan, Wei Du

**Affiliations:** Key Laboratory of Symbolic Computation and Knowledge Engineering of Ministry of Education, College of Computer Science and Technology, Jilin University, Changchun 130012, China; sunhm21@mails.jlu.edu.cn (H.S.); quhw19@mails.jlu.edu.cn (H.Q.); duankf@mails.jlu.edu.cn (K.D.)

**Keywords:** graphical neural networks, multi-view graphs, cell type identification, single-cell RNA sequencing

## Abstract

Single-cell RNA sequencing (scRNA-seq) data reveal the complexity and diversity of cellular ecosystems and molecular interactions in various biomedical research. Hence, identifying cell types from large-scale scRNA-seq data using existing annotations is challenging and requires stable and interpretable methods. However, the current cell type identification methods have limited performance, mainly due to the intrinsic heterogeneity among cell populations and extrinsic differences between datasets. Here, we present a robust graph artificial intelligence model, a multi-view graph convolutional network model (scMGCN) that integrates multiple graph structures from raw scRNA-seq data and applies graph convolutional networks with attention mechanisms to learn cell embeddings and predict cell labels. We evaluate our model on single-dataset, cross-species, and cross-platform experiments and compare it with other state-of-the-art methods. Our results show that scMGCN outperforms the other methods regarding stability, accuracy, and robustness to batch effects. Our main contributions are as follows: Firstly, we introduce multi-view learning and multiple graph construction methods to capture comprehensive cellular information from scRNA-seq data. Secondly, we construct a scMGCN that combines graph convolutional networks with attention mechanisms to extract shared, high-order information from cells. Finally, we demonstrate the effectiveness and superiority of the scMGCN on various datasets.

## 1. Introduction

Single-cell RNA sequencing (scRNA-seq) is a technology that measures the gene expression of individual cells using high-throughput sequencing. It can analyze millions of cells from different biological samples and conditions [1,2,3,4,5]. This technology can reveal the diversity and characteristics of cells and help identify their types [6]. It can also provide new insights into how cells interact and function in complex systems [7,8,9]. Moreover, scRNA-seq can support research in healthcare, drug development, and biotechnology [10,11,12,13]. Single-cell RNA sequencing (scRNA-seq) has enabled powerful and straightforward access to the transcriptomes of individual cells [1]. With the help of scRNA-seq technology, it has become easier to understand individual cell organizations, leading to a better understanding of immunity and various diseases. Cancer is often considered the most heterogeneous and complex of all diseases [14]. The presence of cancer stem cells is a significant source of tumor formation, drug resistance, and metastasis [15]. The early detection of cancer stem cells is crucial to ensuring the adequate diagnosis and treatment of cancer. scRNA-seq aids in detecting genetic information and controlling genes and differences in gene expression within individual cells [16]. Therefore, with the support of scRNA-seq technology, it is easy to understand the heterogeneity within tumors, analyze cancer stem cells, and map clones in the tumor. scRNA-seq can provide valuable information for cancer research. Due to all these characteristics, scRNA-seq is increasingly used in the study of cancers. As scRNA-seq becomes more advanced and affordable, more and more scRNA-seq data are generated and used in various biomedical fields. This differs from traditional methods that analyze the average expression of thousands of cells in a sample, which can miss the variability and details of single cells.

Identifying cell types from single-cell transcriptomics data is an important goal in helping to explain the diversity and complexity of tissues and organisms. There are several methods to obtain single-cell sequencing data, such as Drop-seq [17], inDrop [18], Chromium [19], and smart-seq2 [20]. However, these methods can produce different kinds of noise [21,22] and batch effects [23] in the data, which can affect the accuracy of cell identification. Batch effects can also arise from different platforms [24,25], omics types [26], and species [4,27] in the data. Cross-species single-cell analysis is a new field that can study how cells evolve and develop across different species [28]. However, cross-species data can have more severe batch effects. Reducing these batch effects is a significant challenge for cell identification methods.

One way to analyze scRNA-seq data is to cluster cells into different groups [29]. However, many clustering methods have limitations, such as needing to know the number of groups in advance or taking too much time and memory [30]. Some methods can identify cell types from various scRNA-seq data, such as Seurat (https://satijalab.org/seurat/, accessed on 26 August 2022) [31], Conos (https://www.nature.com/articles/s41592-019-0466-z, accessed on 26 August 2022) [32], scmap (https://www.nature.com/articles/nmeth.4644, accessed on 26 August 2022) [33], and CHETAH (https://academic.oup.com/nar/article/47/16/e95/5521789, accessed on 26 August 2022) [34]. Seurat [31] is a software tool for single-cell RNA sequencing data analysis, which provides functions such as data quality control, single-cell clustering, differential expression gene identification, gene function annotation, and pathway analysis, as well as visualization analysis. In addition, Seurat v3 proposes a cell-type identification method based on anchors that can be applied to different single-cell datasets. Conos [32] is a method for integrating multiple single-cell RNA-seq datasets, focusing on aligning homologous cell types across heterogeneous sample collections. It generates a joint graph representation through pairwise alignments, enabling the propagation of labels from one sample to another. Scmap [33] is a method for projecting cells obtained from scRNA-seq experiments onto cell types identified in different experiments. It learns cell types by measuring the maximum similarity between a reference dataset with good cell annotations and an unknown dataset. CHETAH [34], guided by existing reference data, defines a classification tree for top-down classification in data lacking annotations. Although these methods can be effective in various scenarios, they still have limitations. One problem is that they only use the information from each cell and ignore the connections between cells. Graph convolutional networks (GNNs) are a new way to analyze data, such as cells and their connections, with a graph structure [35]. GNNs can learn the features and relationships of cells and improve the performance of different tasks. Graph convolutional networks (GCNs) are GNNs applied to single cells and diseases [36,37,38,39,40]. GNNs have also been used to analyze scRNA-seq data, such as imputation and clustering [41,42,43]. These results inspire us to use GNNs for cell identification, a new and promising direction.

Most methods for cell identification use the information from each cell, but scGCN [44] uses graph convolutional networks (GCN) to use the connections between cells. This improves the performance of cell identification and shows the potential of graph convolutional networks (GNNs) for this task. Different graph structures can capture different information from scRNA-seq data, but scGCN depends on one graph structure. Multi-view learning can use multiple graph structures to reduce the batch effects from different sequencing methods. In this article, we use a multi-view graph convolutional network model (scMGCN) for cell identification on scRNA-seq data. Figure 1 provides a simplified workflow of our approach.

The main contributions of this article are the following:We use multi-view learning and multiple graph construction methods to create different graph structures from raw scRNA-seq data. We then use graph convolutional networks to learn from these graphs and get complete information about the cells. This also helps us reduce the batch effects from different sequencing methods in the cell identification task.We develop scMGCN, a multi-view graph convolutional network that uses graph convolutional networks and attention mechanisms. scMGCN can learn the common and specific information from each cell and the relationships between cells.Through benchmarking with other state-of-the-art cell type identification methods on single-species, cross-species, and cross-platform datasets, scMGCN consistently demonstrates superior accuracy in different tissues, platforms, and species.

This paper is structured as follows: In the Section 4, we describe the data, the data preprocessing, the graph construction, and the model details. In the Section 2, we report the performance evaluations. In the Discussion section, we discuss the strengths and limitations of our method. In the Conclusions section, we summarize our main findings and contributions.

## 2. Results

### 2.1. Performance of Cell Type Identification on Single Dataset

We compare the performance of scMGCN with the other five methods for cell type identification on single-cell RNA sequencing (scRNA-seq) data from five datasets, which have various challenges in cell type recognition, such as heterogeneity, batch effects, and rare cell types. The compared methods contain Seurat v3 [31], Conos [32], scmap [33], CHETAH [34], and scGCN (https://www.nature.com/articles/s41467-021-24172-y, accessed on 26 August 2022) [44]. We use accuracy as the primary evaluation metric and perform five-fold cross-validation, reporting the average results. We also calculate the F1-macro scores to measure the performance of minority cell types. We find that scGCN outperforms the other methods regarding accuracy and F1-macro scores. The results of both metrics are shown in Table 1 and Appendix A.

Our model achieves the highest mean accuracy (90.61) across all datasets, slightly better than Seurat v3 (89.84) and scGCN (87.44). scMGCN also outperforms Conos (83.85), scmap (77.33), and CHETAH (76.84) by a more significant margin. In a few datasets, such as GSE98638 and SRP073767, scMGCN achieved the second best performance. The F1-macro scores of scMGCN vary more than the other methods across the datasets, but scMGCN still shows good stability in terms of performance. These results demonstrate that scMGCN can effectively integrate multiple graphs and complement the missing information in a single graph for cell type identification.

### 2.2. Cell Type Identification across Datasets of Different Species

We evaluate the performance of scMGCN and other four methods, Seurat v3, Conos, scmap, and scGCN, for cross-species cell type identification on two pairs of human and mouse scRNA-seq datasets. We exclude CHETAH from this experiment because it does not perform data integration across species. We split the training set into a training subset (80%) and a validation subset (20%) within each cell type. We use accuracy and F1-macro scores as the evaluation metrics and perform five-fold cross-validation, reporting the average results.

Table 2 and Appendix A show a performance comparison between our method and the compared methods on the cross-species datasets. The results show that scMGCN achieves the highest mean accuracy (75.12) across both pairs of datasets, much better than Seurat v3 (62.43), Conos (65.57), scmap (62.83), and scGCN (66.84). This indicates that scMGCN can extract and aggregate the shared high-order cell relationships from multi-view graph data. scMGCN also has the highest mean F1-macro score (71.95), indicating its ability to handle rare cell types across species. These results demonstrate the superior performance of scMGCN in cross-species cell type identification.

### 2.3. Cell Type Identification across Datasets of Different Platforms

We perform cross-platform analysis on six human peripheral blood mononuclear cell (PBMC) datasets and four human pancreatic cell datasets from diverse platforms and sequencing methods. We split the data from different platforms into training and testing sets to ensure rigorous evaluation. Moreover, we further divide the validation set in each cluster based on a 20% ratio in the training set. Using the same evaluation criteria, we compare the accuracy and F1-macro scores of scMGCN with other methods on cross-platform datasets.

Table 3 shows the specific comparison results. Our results demonstrate that scMGCN outperforms other methods in identifying cells across different experimental platforms. Specifically, in the PBMC datasets, scMGCN (88.52) achieves the highest mean accuracy, followed by Seurat v3 (83.46), Conos (81.92), scGCN (85.45), and scmap (64.03). ScMGCN performs well even when the training set contains less than 100 cell samples in PBMC (CEL-Seq2 (https://genomebiology.biomedcentral.com/articles/10.1186/s13059-016-0938-8, accessed on 26 August 2022))-PBMC (10× v3 (https://www.10xgenomics.com/, accessed on 26 August 2022)) datasets. From Table 4, we can observe that scMGCN (98.98) also achieves the highest mean accuracy, followed by scGCN (98.87), Conos (43.74), scmap (86.75), and Seurat v3 (98.79). scMGCN also outperforms other methods regarding mean F1 score in the classification of human pancreatic cell datasets. These results indicate that scMGCN can handle batch effects across platforms better than other methods. The detailed results are shown in Appendix A.

### 2.4. Ablation Experiments

To further evaluate the performance of scMGCN, we compare the performance of multi-view fusion and different network modules in different types of experiments.

#### 2.4.1. The Performance of Multi-View Fusion

To validate the advantage of multi-view fusion, we compared the performance of models using multi-view and single-view in the single dataset, cross-species, and cross-platform experiments. The following steps obtained the single-view results: First, the graph structures of the reference dataset and query dataset were obtained using the corresponding graph construction methods. Then, the graph convolutional network was used to train the single-view, and the feature embeddings of cells with different graph construction methods were obtained. Finally, the cell types of the query dataset were predicted by the classification module. The experimental results are shown in Figure 2 and Appendix A. The figures show that the model using multi-view fusion had a consistently higher performance than the model using single-view in the single dataset, cross-species, and cross-platform experiments, and the performance had good stability in different datasets.

#### 2.4.2. The Performance of Different Network Modules

To validate the effectiveness of each module of scMGCN, we compared the performance of scMGCN and two other existing graph neural network-based multi-view fusion models in the single dataset, cross-species, and cross-platform experiments. These models are heterogeneous graph attention networks (HAN) and Multi-Omics Graph Convolutional NETworks (MOGONET). HAN is a model for analyzing heterogeneous graph data, which uses multiple GATs to process the semantic information of different meta-path graphs and then uses the Attention mechanism to fuse different semantics. The structure of this model can be directly applied to multi-view fusion. MOGONET is a model that uses multiple GCNs to extract information from different views, and further uses a view correlation discovery network View Correlation Discovery Network (VCDN) to aggregate them. The experimental results are shown in Figure 3 and Appendix A. From the figures, it can be seen that scMGCN had a consistently higher performance than the other two models in the single dataset, cross-species, and cross-platform experiments, and the performance had good stability in different datasets.

### 2.5. Model Details and Computational Resources

The specific implementation of the scMGCN model consists of three main components: the Graph Convolutional Layer module, the Attention module, and the MLP module. The Graph Convolutional Layer module comprises six GCNs, with each GCN containing three layers of GraphConv and a dropout layer. The Attention module consists of two linear layers and a tanh activation function. Finally, the MLP module comprises a single linear layer. For specific parameters, please refer to the code.

We compared the time and memory consumption of Seurat v3, Conos, scmap, CHETAH, scGCN, and scMGCN on the single dataset on a system with 128GB memory, an i7-13700F (8-core, 24-thread) CPU, and a 4090 GPU. Seurat v3, Conos, scmap, and CHETAH are all executed on the CPU, while scGCN and scMGCN are executed on the GPU. The results are summarized in Table 5.

## 3. Discussion

Some methods are available for identifying cell types from various scRNA-seq data, such as Seurat, Conos, scmap, and CHETAH. However, these methods ignore the higher-order relationships among cells. Graph neural networks (GNNs) are a new method that uses graph structures to analyze data, such as cells and their connections. A representative GNN model is scGCN, which has learned higher-order relationships among cells but is limited by single-graph learning. We propose scMGCN, a model that uses multi-view graph convolutional neural networks to identify cell types in single-cell RNA sequencing data. The model adopts multi-view learning, generates different data views using various graph construction methods, and then uses graph convolutional neural networks to learn node representations. Finally, the attention-based multi-view embedding aggregation layer combines the learned node representations for cell type identification. We conducted comparative experiments on individual datasets and data from different species and platforms. The experimental results show that scMGCN performs well in single-cell identification tasks, especially in cross-species and cross-platform scenarios. The experimental results show that, although scMGCN had a smaller improvement compared with other methods in a few cases, such as the single dataset cell type recognition task compared with the scGCN method, in most cases, the scMGCN method achieved a better performance compared with other methods.

As an efficient cell type identification method, scMGCN will have widespread applications. Efficient cell type identification techniques are crucial in clinical diagnosis, cell development and differentiation research, and drug development efforts. For example, we can extend scMGCN to the diagnosis of blood diseases. By using scMGCN to classify blood cells from patients, we can determine their pathological types and provide a basis for clinical treatment. We can also use scMGCN to conduct in-depth research on specific cell types, helping researchers understand the role of different cell types in disease development and progression and thus providing new targets and strategies for drug development. In summary, the proposed cell type identification model, scMGCN, has broad and profound applications in medicine, biology, and related fields. It is of great significance for understanding the processes of life and preventing and treating diseases.

Although our method achieved good results on different datasets, there are still some limitations to our approach. Regarding dataset quality, our method is still influenced by the reference dataset. The quality of the reference dataset directly affects the effectiveness of our final model, so in the future, we can introduce some preprocessing methods for the dataset to improve its quality. Secondly, our method currently only processes single-cell RNA-seq-related data. Our model achieved different results on different single-species, cross-species, and cross-platform datasets, demonstrating high sensitivity to different datasets. In the future, we can further adjust the relevant parameters and the number of network layers in the model to overcome our model’s sensitivity to data variations. Regarding data learning and interpretation, our model adopts an attention-based multi-view aggregation method to learn and interpret data from different perspectives. Although this improves the performance of our model, scMGCN currently does not assign explicit edge weights to the relationships between units in each graph. It only reflects correlation but cannot reflect the strength of relationships. In the future, the introduction of edge weights may be considered. However, it is necessary to consider how to handle the differences in edge weight definitions between different graph construction methods.

Finally, with technological advancements, the analysis of single-cell RNA sequencing data (scRNA-seq) also faces challenges. Firstly, the amount of data generated by scRNA-seq technology is vast, and there is much noise. Secondly, analyzing cell heterogeneity, incredibly accurately identifying and distinguishing different cell subpopulations, remains challenging. Furthermore, differences in experimental conditions, platforms, and batches make it difficult to compare different datasets directly. In this context, new algorithms and technologies like the scMGCN model provide new opportunities and possibilities for addressing these challenges. scMGCN utilizes different graph construction methods and graph convolutional networks to analyze single-cell data, enabling it to capture the similarities and differences among cells better and to be used for cell type identification and classification. However, despite the many advantages of the scMGCN model, it still faces some challenges. An example of a challenge in machine learning is improving the model’s generalization ability to handle more extensive and higher-dimensional data. Another challenge is integrating the model with other advanced machine learning techniques to enhance the accuracy and reliability of cell type identification.

## 4. Materials and Methods

### 4.1. Data Collection

The rapid development of single-cell technologies has led to a significant increase in single-cell omics data. As more single-cell datasets become available, there is an urgent need to leverage existing and newly generated data in a reliable and reproducible manner, learn from well-established single-cell datasets with clearly defined labels, and transfer these labels to newly generated datasets to assign cell-level annotations. However, existing and newly generated datasets are often collected from different tissues and species under various experimental conditions, through different platforms, and across different omics types. Therefore, to meet the demands of practical applications, we conducted three types of experiments to evaluate the performance of scMGCN on individual datasets and datasets from different species and platforms. These datasets represent different scenarios and challenges for cell label transfer. To highlight the comparison of specific cell types, we evaluated the performance of scMGCN on individual datasets. Next, we evaluated the performance of scMGCN on datasets from different species. Since different experimental platforms generate emerging single-cell datasets, we tested whether scMGCN can accurately transfer labels between datasets from different platforms.

All datasets were obtained from public databases, as shown in Table 6. For evaluating the performance of the single-dataset experiment, we used five datasets as follows: the dataset GSE115746 (https://www.ncbi.nlm.nih.gov/geo/query/acc.cgi?acc=GSE115746, accessed on 26 August 2022), which contains 9035 cells from the mouse anterior lateral motor cortex (ALM); the dataset PHS001790, which contains 12,552 human cells from the middle temporal gyrus; the dataset GSE118389 (https://www.ncbi.nlm.nih.gov/geo/query/acc.cgi?acc=GSE118389, accessed on 26 August 2022), which contains 1534 cells from six human triple-negative breast cancer (TNBC) tumors; the dataset GSE72056 (https://www.ncbi.nlm.nih.gov/geo/query/acc.cgi?acc=GSE72056, accessed on 26 August 2022), which contains 4645 human melanoma cells; the dataset GSE98638 (https://www.ncbi.nlm.nih.gov/geo/query/acc.cgi?acc=GSE98638, accessed on 26 August 2022), which contains 5063 T cells from patients with hepatocellular carcinoma; the dataset GSE85241 (https://www.ncbi.nlm.nih.gov/geo/query/acc.cgi?acc=GSE85241, accessed on 26 August 2022), which contains 2122 pancreatic cells from human cadavers; the dataset GSE109774 (https://www.ncbi.nlm.nih.gov/geo/query/acc.cgi?acc=GSE109774, accessed on 26 August 2022), which contains 54,865 single cells from 20 tissues of 3-month-old mice; the dataset SRP073767 (https://www.ncbi.nlm.nih.gov/Traces/index.html?view=study&acc=SRP073767, accessed on 26 August 2022), which contains 65,943 human peripheral blood mononuclear cells; and the dataset GSE120221 (https://www.ncbi.nlm.nih.gov/geo/query/acc.cgi?acc=GSE120221, accessed on 26 August 2022), which contains 10,495 bone marrow mononuclear cells. We used two pairs of data sets to evaluate the performance across data from different species. The first data pair is from the GSE115746 (https://www.ncbi.nlm.nih.gov/geo/query/acc.cgi?acc=GSE115746, accessed on 26 August 2022) and PHS001790 (https://www.ncbi.nlm.nih.gov/projects/gap/cgi-bin/study.cgi?study_id=phs001790.v2.p1, accessed on 26 August 2022) datasets, which contain 9035 mouse and 12,552 human brain cells. The second data pair is from the GSE120221 (https://www.ncbi.nlm.nih.gov/geo/query/acc.cgi?acc=GSE120221, accessed on 26 August 2022) and the GSE107727 (https://www.ncbi.nlm.nih.gov/geo/query/acc.cgi?acc=GSE107727, accessed on 26 August 2022), which contain 10,495 human bone marrow mononuclear cells and 30,494 mouse bone marrow mononuclear cells. We used one species as the training set and the other as the testing set for each pair. We used two data types to evaluate the performance across data from different platforms. One type is human peripheral blood mononuclear cell (PBMC) data, which are sequenced by different methods: PBMC (10× v2), PBMC (Smartseq2), PBMC (InDrop), PBMC (10× v3), PBMC (CEL-Seq2), and PBMC (Seqwell). The six PBMC data are available from the Broad Institute Single Cell portal (https://portals.broadinstitute.org/single_cell/study/SCP424/single-cell-comparison-pbmc-data, accessed on 26 August 2022) and the Zenodo repository (https://zenodo.org/, accessed on 26 August 2022). The other is human pancreatic cell data, which are also sequenced by different methods: GSE85241 (CEL-Seq2) (https://www.ncbi.nlm.nih.gov/geo/query/acc.cgi?acc=GSE85241, accessed on 26 August 2022), GSE81608 (Smart-seq2) (https://www.ncbi.nlm.nih.gov/geo/query/acc.cgi?acc=GSE81608, accessed on 26 August 2022), and E-MTAB-5061 (Smart-seq2) (https://www.ebi.ac.uk/arrayexpress/experiments/E-MTAB-5061, accessed on 26 August 2022). We choose these single-cell datasets because they are often used to evaluate the performance of cell type identification through different methods [31,33,44,45,46].

### 4.2. Data Preprocessing

We need to preprocess the data to construct graphs and use models from sequencing data to reduce noise. We followed these steps for preprocessing. Firstly, we removed doublet cells in droplet-based methods like drop-seq and inDrop. Doublets are cases where more than one cell is in a droplet. We used Scanpy to calculate and filter out doublet scores. Next, we only kept genes with at least one read and matched the reference and query sets. This made sure that both sets had the same gene dimensions. Then, we selected highly variable genes using ANOVA for multi-category differential analysis. This helped us find the most variable genes across different cell types. We used Bonferroni correction to choose 2000 genes in the reference set with the lowest adjusted *p*-values. We removed all non-variable genes in both sets.

### 4.3. Graph Construction

After data preprocessing, we constructed graphs using different methods to calculate cell relationships, build adjacency matrices, and generate mixed graph adjacency matrices for use as inputs in graph convolutional layers. We used the Scanpy library to normalize total counts for each cell, ensuring that each cell had the same total count after normalization, which helped to minimize the error in similarity calculations due to cell heterogeneity.

In this study, we used six graph construction methods for multi-view aggregation, representing six edges in the multi-view graph. These methods included Approximate Nearest Neighbors Oh Yeah (ANNOY), CCA_MNN, Harmony, Scanorama, Scmap, and Autoencoder-KNN. The relevant details of these six graph construction methods are shown in Appendix A. Each method first constructed a data inter-graph $g_{inter}$ containing the cell relationships within the reference and query datasets, followed by constructing a data intra-graph $g_{intra}$ containing the cell relationships within the query dataset. Both graphs were represented as adjacency matrices, $A^{intra}\in R^{\left(n_{q}\times n_{q}\right)}$ and $A^{inter}\in R^{\left(n_{q}\times n_{q}\right)}$, where $n_{r}$ and $n_{q}$ are the numbers of cells in the reference and query datasets, respectively. These two graphs were combined to form an adjacency matrix $A^{H}\in R^{\left(n_{r}+n_{q}\right)\times \left(n_{r}+n_{q}\right)}$, where $A_{ij}^{H}$ represented the relationship between the *i*th and *j*th cells, with $A_{ij}^{H}$=0 indicating the absence of a relationship. In addition to the adjacency matrix, we used a feature matrix X= $\left[\begin{array}{c} X_{R}\\ X_{Q} \end{array}\right]$ ∈$R^{(n_{r}+n_{q})\times m}$, composed of the preprocessed reference and query matrices, as the starting input features for the graph convolutional layer. Each row represented the starting feature vector of the corresponding node. The starting feature matrix input was the same for all graph construction methods.

### 4.4. Related Comparative Methods

We compared five commonly used methods for cell type identification based on single-cell RNA sequencing data, including Seurat v3 [31], Conos [32], scmap [33], CHETAH [34], and scGCN [44]. Seurat v3 is a widely used and well-validated toolkit for single-cell genomics. Recently, Seurat v3 introduced an anchor-based label transfer method that has broad applicability and can be used with various single-cell samples. Conos utilizes the pairwise arrangement of samples to construct a joint graph representation, allowing label information to be transferred from one sample to another. Scmap learns and infers the cell types in an unknown dataset by comparing its maximum similarity with a reference dataset that has good cell annotations. With the guidance of reference data, CHETAH constructs a classification tree for the top-down classification of unannotated data. scGCN utilizes graph convolutional neural networks to learn complex relationships between cells from constructed graph data in order to complete the task of cell type recognition.

### 4.5. Model Design and Model Training

In this paper, we utilized a graph convolutional neural network for feature learning and employed multiple graph construction methods to describe the same cell data. By leveraging different graphs that capture information from diverse perspectives and can be represented as distinct types of edges, we consider the entire model as a multi-view graph convolutional network. To aggregate different embedding information, we adopt an attention mechanism.

#### 4.5.1. Input Data for scMGCN

For the training of the graph convolutional layer, the $g_{i},i\in \left\{1,2,\ldots ,P\right\}$, constructed by different graph methods, are used as inputs, where *P* corresponds to the number of graph construction methods and views. Each input graph comprises the adjacency matrix and feature matrix, obtained as previously described, with the initial feature matrix input being identical.

#### 4.5.2. Graph Convolutional Layer of scMGCN

During training, the graph-structured data are fed into the graph convolutional layer, where nodes receive information from their neighbors and update their representation, leading to learning the nodes’ underlying features. Graphs generated by different graph construction methods are separately fed into the graph convolutional layer, resulting in corresponding node embedding information. The initial layer of a Graph Convolutional Neural Network (GCN) is intended to accept graph data that have undergone preprocessing, and the processing procedure is shown in Equation (Equation 1):(1)$$H^{\left(1\right)}=f\left(X,\tilde{A}\right)=\sigma \left(\tilde{A}XW^{\left(0\right)}\right)$$

The output of the first layer graph convolution is represented by $H^{\left(1\right)}$, which is obtained by applying a non-linear activation function σ to the product of the weight matrix $W^{\left(0\right)}$ and the input feature matrix *X*. During training, the weight matrix $W^{\left(0\right)}$ is updated using stochastic gradient descent. $\tilde{A}$ is the modification of the adjacency matrix for efficient training in the Graph Convolutional Network (GCN). The specific modification method is shown in Equation (Equation 2):(2)$$\tilde{A}=D^{-\frac{1}{2}}\hat{A}D^{-\frac{1}{2}}=D^{-\frac{1}{2}}\left(A^{H}+I\right)D^{-\frac{1}{2}}\;$$

In Equation (Equation 2), $A^{H}$ represents the adjacency matrix of the original input graph, a square matrix with dimensions equal to the sum of the number of cells in the reference set and query set. *I* denotes the identity matrix with dimensions identical to the adjacency matrix. It represents the one-hot encoded vectors of all nodes in the graph. *D* is the diagonal angle matrix of $\hat{A}$. As the number of layers in Graph Convolutional Neural Networks (GCN) increases, the model learns higher-order neighbor information and aggregates it into node representations. The specific process is shown in Equation (Equation 3):(3)$$H^{\left(l+1\right)}=f\left(H^{\left(l\right)},\tilde{A}\right)=\sigma \left(\tilde{A}H^{\left(l\right)}W^{\left(l\right)}\right)\;$$

Here, *l* represents the number of graph convolutional layers, and $H^{l}$ represents the node feature representation output by the *l*th layer, specifically $H^{\left(0\right)}$=*X*. As the number of graph convolutional layers increases, the learned node embedding becomes more abstract. However, our experiments show no significant improvement in the experimental results when the number of graph convolutional layers exceeds two. On the contrary, it increases the time cost of the program and causes overfitting problems in some datasets. Hence, our experiments are based on a two-layer GCN.

#### 4.5.3. Multi-View Aggregation of scMGCN

Next, we need to fuse the information generated by different graphs. In this paper, we propose an attention-based method to accomplish this task. We take *P* sets of node embedding information $\left(E_{1},\ldots ,E_{P}\right)$, learned in the graph convolutional layers, as input and obtain the weights $\left(\beta_{1},\ldots ,\beta_{P}\right)$ for different node embedding information using Equation (Equation 4):(4)$$\left(\beta_{1},\ldots ,\beta_{P}\right)=att\left(E_{1},\ldots ,E_{P}\right)\;$$

In Equation (Equation 4), *P* represents the number of graph construction methods and the number of views and types of edges. The function att(·) denotes a deep neural network capable of performing the attention mechanism. $E\in R^{(n_{r}+n_{q})\times h}$ represents the output of the final layer of the graph convolutional layers, where *h* is the output feature dimension of the graph convolutional layers. To learn the importance of each graph, we first perform nonlinear transformations on different embedding information. We then calculate the importance of different embedding information $w_{i},$i∈{1,2,…,P} by measuring the similarity between the transformed embedding information and the attention vector. Additionally, the importance of embedding information for all nodes is averaged. The specific process is shown in Equation (Equation 5):(5)$$w_{p}=\frac{1}{v}\sum_{i\in v}q^{T}\cdot \tanh\left(W\cdot e_{i}^{p}+b\right)\;$$

Here, *v* denotes the set of all nodes in graph p, *W* represents the weight matrix, *b* represents the bias vector, *q* represents the attention vector used to calculate similarity, and $e_{i}^{p}\in E_{P}$ represents the embedding information after nonlinear transformation. In order to enable a fair comparison, the variables mentioned above are shared among various node embeddings. We use softmax to normalize the importance of embedding information for all groups of nodes and obtain the weight of different node embeddings, representing the weight of different graph construction methods. The specific process is shown in Equation (Equation 6):(6)$$\beta_{i}=\frac{\exp(w_{p})}{\sum_{p=1}^{P}\exp\left(w_{p}\right)}$$

Here, β represents the weight of different graph construction methods. The higher the β, the more critical the corresponding graph construction method. The weights of different graph construction methods also vary for different types of tasks. Using these weights as coefficients, we can fuse the node embedding information of different graph construction methods to obtain the final embedding result *E*. The specific process is shown in Equation (Equation 7):(7)$$E=\sum_{p=1}^{P}\beta_{p}\cdot e^{p}\;$$

#### 4.5.4. Result Output of scMGCN

As shown in Equation (Equation 8), after aggregating the node embedding information of multiple views, we use a multilayer perception (MLP) to obtain the predicted labels.
(8)$$\tilde{y_{i}}=MLP\left(E_{i}\right)$$

We calculate the cross-entropy between the predicted and true cell types and minimize the cross-entropy to calculate the model loss. The loss function of the model is shown in Equation (Equation 9):(9)$$L_{c}=-\sum_{l\in y_{L}}Y^{l}\ln\left(C\cdot E^{l}\right)$$

Among them, *C* is the parameter of the Multilayer Perceptron classifier, $y_{L}$ is the index set of labeled nodes, and $Y^{l}$ is the label of labeled nodes, which is also the cell type. $E^{l}$ is the final node embedding of labeled nodes. We can optimize the model and learn the node embedding by utilizing backpropagation. The multi-view aggregation output is the embedding vector of nodes, so the loss function is highly adaptable. It can be customized for different tasks, such as link prediction.

The scMGCN model adopts a multi-view approach to transform the problem into a multi-view graph convolutional network. This allows us to integrate cell relationships from different perspectives and improve the accuracy of cell recognition. The result is a more stable and effective cell recognition system.

### 4.6. Performance Metrics

The model predicts labels for cells, which represent the cell types. In this paper, the primary metrics used to assess the model’s performance in predicting cell labels are the $F_{1}$ score and accuracy. Accuracy refers to the proportion of correctly predicted samples out of the total samples.
(10)$$Precision=\frac{TP}{TP+FP}$$
(11)$$Recall=\frac{TP}{TP+FN}$$

Equation (Equation 10) represents the calculation process for precision, while Equation (Equation 11) represents the calculation process for recall. In this context, FP stands for false positives, FN represents false negatives, and TP denotes true positives. In general classification tasks, Precision and Recall are calculated for each sample separately based on the cell type, while accuracy is calculated across all samples. The calculation process of $F_{1}$ score is shown in Equation (Equation 12):(12)$$F_{1}=2\cdot \frac{Precision\cdot Recall}{Precision+Recall}$$

Regarding cell type identification, each task is a binary classification task. To calculate the overall $F_{1}$ score, the $F_{1}$ scores of negative and positive samples need to be combined. The sub-library metrics in the sklearn module of Python provide two different ways of combining, namely macro and micro. Macro first calculates the precision and recall by cell type, then takes the average of all $F_{1}$ scores. This approach can reflect the model’s performance in predicting cell types that comprise a small proportion of the total population. Micro, on the other hand, does not explicitly distinguish between types when calculating.

## 5. Conclusions

We propose a novel approach that uses multi-view graph convolutional neural networks and multi-view learning to identify cell types in single-cell RNA sequencing data. Our method consists of four steps. First, we construct graphs from single-cell transcriptomics data using different graph construction methods. Second, we pass the graphs through different graph convolutional layers to obtain different node embeddings. Third, we fuse the node embeddings using an attention mechanism to obtain new cell node embeddings. Fourth, we input the new cell node embeddings into a multi-layer perceptron-based classifier for cell type classification. Our experiments on single-species, cross-species, and cross-platform datasets show that our model can achieve good classification results.

## Figures and Tables

**Figure 1 ijms-25-02234-f001:**
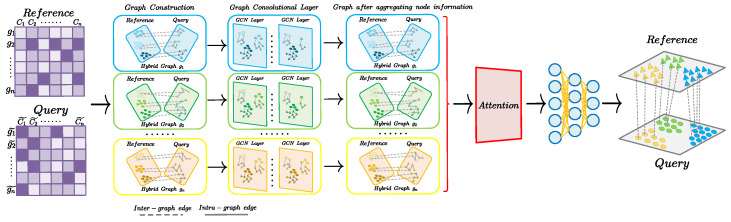
Architecture diagram of scMGCN for cell type identification.

**Figure 2 ijms-25-02234-f002:**
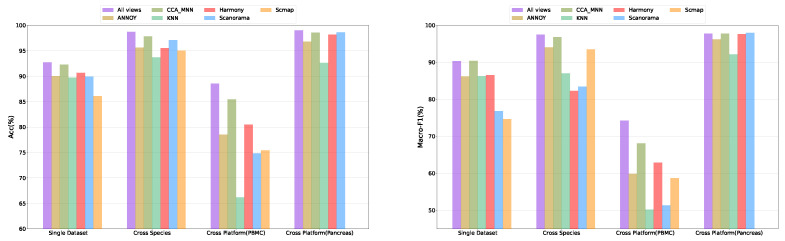
Mean performance of multi-view and single-view methods in single dataset, cross-species, and cross-platform experiments with respect to Accuracy and Macro-F1.

**Figure 3 ijms-25-02234-f003:**
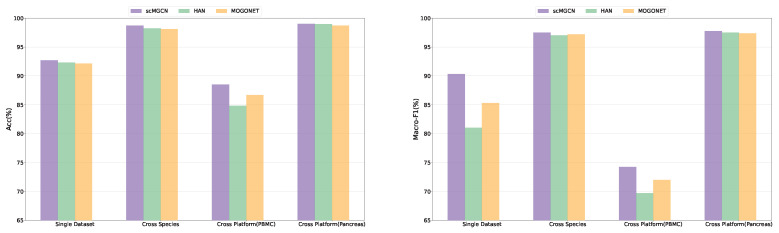
Mean performance comparison of scMGCN and two other existing graph neural network-based multi-view fusion models in single dataset, cross-species, and cross-platform experiments with respect to Accuracy and Macro-F1.

**Table 1 ijms-25-02234-t001:** Accuracy and F1 score of cell identification task for single dataset.

	Seurat v3		Conos		scmap		CHETAH		scGCN		scMGCN	
	Accuracy	F1	Accuracy	F1	Accuracy	F1	Accuracy	F1	Accuracy	F1	Accuracy	F1
GSE115746	**100**	**100**	**100**	**100**	**100**	**100**	96.27	56.72	99.93	95.81	99.94	95.81
GSE118389	93.97	89.98	93.97	89.08	88.16	69.68	73.21	49.06	94.19	90.21	**96.43**	**90.63**
GSE72056	92.15	87.30	82.80	78.48	72.00	54.41	90.03	89.54	92.74	89.44	**93.61**	**90.79**
GSE98638	73.16	74.50	70.21	70.76	37.71	34.81	62.53	35.91	**74.49**	**76.83**	73.55	75.01
PHS001790	99.82	97.70	99.94	**99.95**	99.66	98.55	97.38	58.35	99.94	99.67	**99.95**	99.39
GSE85241	98.87	98.38	98.56	98.08	97.60	97.40	96.47	55.14	97.91	97.06	**99.68**	**99.39**
GSE109774	93.25	70.25	73.99	47.27	92.25	77.26	70.13	39.09	89.25	51.63	**94.20**	**79.77**
SRP073767	**66.38**	**53.90**	64.36	50.52	49.71	43.79	28.61	17.32	63.35	40.21	66.12	51.15
GSE120221	91.01	89.91	70.80	65.74	58.92	49.78	76.95	41.36	75.20	45.67	**92.03**	**90.37**
Mean	89.84	84.66	83.85	77.76	77.33	69.52	76.84	49.17	87.44	76.28	**90.61**	**85.81**

For each dataset, we split the cells into training (60%) and test (40%) sets according to their true labels. We further divide the training set into a training subset (80%) and a validation subset (20%) within each cell type. Then, we perform five-fold cross-validation. All of the above results are derived from the test set. The bold text in each row of the table indicates the maximum value among all the data in that specific row.

**Table 2 ijms-25-02234-t002:** Accuracy and F1 score of cell identification task for cross species datasets.

	Seurat v3		Conos		scmap		scGCN		scMGCN	
	Accuracy	F1	Accuracy	F1	Accuracy	F1	Accuracy	F1	Accuracy	F1
PHS001790 (human)–GSE115746 (mouse)	**100**	**100**	**100**	**100**	99.71	84.92	99.76	98.33	99.90	99.22
GSE115746 (mouse)–PHS001790 (human)	72.94	74.78	97.37	71.97	97.29	70.74	99.92	99.48	**99.98**	**98.40**
GSE120221 (human)–GSE107727 (mouse)	40.33	36.95	37.39	37.75	33.75	21.01	29.44	30.57	**49.69**	**45.92**
GSE107727 (mouse)–GSE120221 (human)	36.44	17.82	27.51	18.85	20.57	20.57	38.24	28.76	**53.91**	**44.26**
Mean	62.43	57.39	65.57	57.14	62.83	49.31	66.84	64.29	**75.12**	**71.95**

For each dataset, we split the cells into training (60%) and test (40%) sets according to their true labels. We further divide the training set into a training subset (80%) and a validation subset (20%) within each cell type.Then, we perform five-fold cross-validation. All of the above results are derived from the test set. The bold text in each row of the table indicates the maximum value among all the data in that specific row.

**Table 3 ijms-25-02234-t003:** Accuracy and F1 score of cell identification task for PBMC datasets.

	Seurat v3		Conos		scmap		scGCN		scMGCN	
	Accuracy	F1	Accuracy	F1	Accuracy	F1	Accuracy	F1	Accuracy	F1
PBMC(10× v2)-PBMC(Smartseq2)	84.58	63.58	84.98	61.93	44.26	34.33	87.35	**71.29**	**90.12**	70.52
PBMC(CEL-Seq2)-PBMC(10× v3)	79.29	57.58	79.86	60.27	79.39	61.62	86.90	73.62	**89.70**	**74.28**
PBMC(InDrop)-PBMC(10× v3)	81.28	59.60	79.08	58.57	70.91	51.93	86.13	64.39	**90.50**	**76.85**
PBMC(Seqwell)-PBMC(Smartseq2)	**92.49**	59.52	87.35	57.10	44.66	37.69	82.60	53.04	86.96	**72.61**
PBMC(10× v3)-PBMC(InDrop)	79.70	54.17	78.36	67.42	80.94	68.93	83.42	69.31	**85.32**	**76.86**
Mean	83.46	58.89	81.92	61.05	64.03	50.90	85.45	68.07	**88.52**	**74.22**

For each dataset, we split the cells into training (60%) and test (40%) sets according to their true labels. We further divide the training set into a training subset (80%) and a validation subset (20%) within each cell type. Then, we perform five-fold cross-validation. All of the above results are derived from the test set. The bold text in each row of the table indicates the maximum value among all the data in that specific row.

**Table 4 ijms-25-02234-t004:** Accuracy and F1 score of cell identification task for human pancreatic cell datasets.

	Seurat v3		Conos		scmap		CHETAH		scGCN		scMGCN	
	Accuracy	F1	Accuracy	F1	Accuracy	F1	Accuracy	F1	Accuracy	F1	Accuracy	F1
GSE81608-GSE85241	97.17	96.41	44.07	27.40	96.85	95.54	47.36	32.28	97.87	97.06	**98.01**	**97.18**
GSE85241-GSE81608	**99.65**	**98.76**	40.57	23.82	73.56	46.14	18.35	29.83	99.23	98.60	99.10	97.44
E-MTAB-5061-GSE85241	97.55	97.12	43.05	29.06	96.65	94.00	95.23	54.90	98.06	97.40	**98.20**	**97.23**
GSE85241-E-MTAB-5061	**99.51**	**99.43**	39.16	24.75	67.91	39.51	0.21	0.51	98.88	98.54	99.31	98.96
GSE81608-E-MTAB-5061	99.23	98.89	47.70	27.08	86.31	71.38	37.70	30.62	99.37	99.21	**99.51**	**99.36**
E-MTAB-5061-GSE81608	99.65	98.90	47.89	28.18	99.24	97.56	98.61	56.13	**99.79**	99.50	99.72	**99.72**
Mean	98.79	98.25	43.74	26.72	86.75	74.02	49.58	34.05	98.87	98.39	**98.98**	**98.32**

For each dataset, we split the cells into training (60%) and test (40%) sets according to their true labels. We further divide the training set into a training subset (80%) and a validation subset (20%) within each cell type. Then, we perform five-fold cross-validation. All of the above results are derived from the test set. The bold text in each row of the table indicates the maximum value among all the data in that specific row.

**Table 5 ijms-25-02234-t005:** Memory consumption and Time of cell identification task for single dataset.

	Seurat v3		Conos		scmap		CHETAH		scGCN		scMGCN	
	Memory	Time	Memory	Time	Memory	Time	Memory	Time	Memory	Time	Memory	Time
GSE115746	1944 M	56.37 s	2105 M	50.76 s	1668 M	15.28 s	1503 M	46.63 s	6042 M	929.82 s	3626 M	6.62 s
GSE118389	940 M	5.71 s	1014 M	8.36 s	1180 M	2.71 s	1423 M	37.05 s	1450 M	46.66 s	2278 M	5.66 s
GSE72056	1218 M	23.24 s	1423 M	29.74 s	1620 M	9.24 s	1919 M	37.81 s	2970 M	276.4 s	2894 M	11.44 s
GSE98638	1478 M	28.11 s	1616 M	34.01 s	1480 M	13.50 s	1439 M	37.34 s	2970 M	238.87 s	2980 M	32.50 s
PHS001790	2640 M	119.89 s	2183 M	103.92 s	2188 M	41.33 s	1628 M	50.34 s	9114 M	1291.09 s	4730 M	9.20 s

**Table 6 ijms-25-02234-t006:** The detailed information of various datasets for different experiment types.

Dataset	Tissue	Cell Number	Protocol	Sample Count
GSE81608	Human Pancreas	1449	SMARTer	1600
GSE85241	Human Pancreas	2122	CEL-Seq2	32
E-MTAB-5061	Human Pancreas	2133	Smart-Seq2	3514
GSE115746	Mouse Brain	12,832	Smart-Seq2	140
GSE109774	Mouse	54,865	10× Genomics	46
SRP073767	Human PBMC	65,943	10× Genomics	29
GSE108989	Human PBMC	8530	Smart-Seq2	12
GSE120221	Human Bone_Marrow	10,495	10× Genomics v2	25
GSE107727	Mouse Bone_Marrow	30,494	10× Genomics v2	8
PHS001790	Human Brain	12,552	Smart-seq2	8
GSE118389	Human Breast	1534	Smart-seq2	1534
GSE72056	Human Melanoma	4645	Smart-seq2	4645
GSE98638	Human Liver	5063	Smart-seq2	6
PBMC	Human PBMC	3222	10× Genomics v2	3222
PBMC	Human PBMC	253	Smart-seq2	253
PBMC	Human PBMC	3222	inDrop	3222
PBMC	Human PBMC	3222	10× Genomics v3	3222
PBMC	Human PBMC	253	CEL-Seq2	253
PBMC	Human PBMC	3176	Seqwell	3176

## Data Availability

The relevant data can be accessed at the following URL: https://www.ncbi.nlm.nih.gov/ (accessed on 26 August 2022); https://www.omicsdi.org/ (accessed on 26 August 2022); https://portals.broadinstitute.org/single_cell/study/SCP424/single-cell-comparison-pbmc-data (accessed on 26 August 2022); https://zenodo.org/ (accessed on 26 August 2022); https://portal.brain-map.org/atlases-and-data/rnaseq (accessed on 26 August 2022).

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
