# Peer review of "scMGCN: A Multi-View Graph Convolutional Network for Cell Type Identification in scRNA-seq Data"

_ijms, 2024, doi:10.3390/ijms25042234_

Round 1

Reviewer 1 Report

Comments and Suggestions for Authors

The paper addresses the increasingly relevant challenge of identifying cell types from complex single-cell RNA sequencing (scRNA-seq) data using a robust and interpretable methodology. The authors introduce scMGCN, a multi-view graph convolutional network model that ingeniously combines multiple graph structures derived from raw scRNA-seq data. The model's integration of graph convolutional networks with attention mechanisms presents an innovative approach to learning cell embeddings and predicting cell labels.

Overall, the paper offers a valuable contribution to the field by introducing an advanced scRNA-seq cell type identification model, scMGCN, demonstrating superior performance in diverse experimental settings.

Comments on the Quality of English Language

Minor editing of English language required

Author Response

Thank you very much for your valuable and meaningful comments. These comments have provided great help for our revision process. We appreciate your patience and continuous help.

Reviewer#1,Concern#1:“The paper addresses the increasingly relevant challenge of identifying cell types from complex single-cell RNA sequencing (scRNA-seq) data using a robust and interpretable methodology. The authors introduce scMGCN, a multi-view graph convolutional network model that ingeniously combines multiple graph structures derived from raw scRNA-seq data. The model's integration of graph convolutional networks with attention mechanisms presents an innovative approach to learning cell embeddings and predicting cell labels. Overall, the paper offers a valuable contribution to the field by introducing an advanced scRNA-seq cell type identification model, scMGCN, demonstrating superior performance in diverse experimental settings. Minor editing of English language required.”

Author response: Thank you for your recognition of our work. We have made the necessary textual modifications.

Author action: We have further revised and refined the English sentences in the manuscript.

Reviewer 2 Report

Comments and Suggestions for Authors

The authors describe an innovative method, leveraging a multi-view graph convolutional network model , to assign cell identities in scRNA-seq datasets. The results presented highlight an overall good performance by the presented tool, which consistently achieves better results when compared to existing state of the art tools, in particular in cross-species comparisons.

However, there are two major weaknesses that must be addressed to allow publication:

1) The authors should make available the software (as a compiled standalone or source code) for testing describing clearly the hardware requirements and software dependencies. A minimal manual and sample dataset is also required. Publishing source code is not strictly required, however the authors may take into account that this is the usual practice in the field.

2) the authors should discuss the performance also in terms of computational resources required by their algorithm, running a set of benchmark tests to document the RAM memory and disk space requirements of scMGCN and the CPU time required by scMGCN and by the other tools tested (seurat v3, CONOS, scMAP etc) to analyse the datasets used in the manuscript. The authors should also describe the hardware used for benchmarking and comment on these results.

minor points:

lines 153 and 168, the use of the term "significantly" is misleading as no statistical test was performed. I suggest replacing it with "consistently".

Author Response

Thank you very much for your valuable and meaningful comments. These comments have provided great help for our revision process. We appreciate your patience and continuous help.

Reviewer#2, Concern#1:“The authors should make available the software (as a compiled standalone or source code) for testing describing clearly the hardware requirements and software dependencies. A minimal manual and sample dataset is also required. Publishing source code is not strictly required, however the authors may take into account that this is the usual practice in the field.”

Author response: I am extremely grateful for your reminder. We have uploaded the relevant code of our model to the GitHub link, and provided a detailed description on how to use the code in the accompanying documentation. Additionally, we have uploaded the required dataset for our model, making it available for readers' reference and use. Thank you again for your assistance.

Author action: We have uploaded our model's relevant code and the associated dataset to the GitHub website, along with detailed instructions on how to use the code. Thank you for your assistance.

Reviewer#2, Concern#2:“The authors should discuss the performance also in terms of computational resources required by their algorithm, running a set of benchmark tests to document the RAM memory and disk space requirements of scMGCN and the CPU time required by scMGCN and by the other tools tested (seurat v3, CONOS, scMAP etc) to analyse the datasets used in the manuscript. The authors should also describe the hardware used for benchmarking and comment on these results.”

Author response: Thanks. This is needed.

Author action: We conducted tests to assess the disk space requirements and CPU time for processing single datasets using Seurat v3, Conos, scmap, and CHETAH, as well as the disk space requirements and GPU time for processing single datasets using scMGCN. Detailed descriptions of the relevant data and the hardware used for testing are provided in the Supplementary Materials.

Reviewer#2, Concern#3: “lines 153 and 168, the use of the term ‘significantly’ is misleading as no statistical test was performed. I suggest replacing it with ‘consistently’.”

Author response: Thank you for your guidance. We have made the relevant changes according to your suggestions.

Author action: We changed "significantly" in line 153 and line 168 of the original manuscript to "consistently" according to the comment.

Reviewer 3 Report

Comments and Suggestions for Authors

The manuscript written by Sun et al. presents a convolutional network method to identify cell types in single-cell RNA-Seq studies. However, the authors used a dataset with only 4 human samples and 2 mouse samples for the validation of their method (GSE84133). A more robust assay should be used for the analyses.

Other suggestions:

1. The authors intend to present their method as more effective in single-cell identification. However, they do not provide the advantages/limitations of the more commonly used methods. For readers unfamiliar with the methodology, this information is relevant to understanding the manuscript results. Also, the methods used in comparison are not presented in the Materials and Methods section.

2. The second paragraph of the Introduction section presents many inherent limitations of the RNA-seq and/or single-cell technologies, not exclusive from the analysis. Since the authors propose to present an alternative to the analysis, not for the technology itself, I believe the analysis limitations should be better detailed instead of the technology ones.

3. What were the criteria for the dataset selection? GSE84133, as I mentioned, has only 4 human samples and only 2 mouse samples. GSE115746 is from the cortex, whilst GSE84133 is from pancreatic islets. What is the biological context behind tissue selection?

Author Response

Thank you very much for your valuable and meaningful comments. These comments have provided great help for our revision process. We appreciate your patience and continuous help.

Reviewer#3, Concern#1:The manuscript written by Sun et al. presents a convolutional network method to identify cell types in single-cell RNA-Seq studies. However, the authors used a dataset with only 4 human samples and 2 mouse samples for the validation of their method (GSE84133). A more robust assay should be used for the analyses.

Author response: Thank you for your feedback. In our experiments, we utilized not only the GSE84133 dataset. We employed three distinct types of datasets: single species dataset, cross species datasets, and cross platform datasets. For the single dataset, we utilized GSE115746, PHS001790, GSE118389, GSE72056, and GSE98638. For the cross species datasets, we used GSE84133(mouse), GSE84133(human), GSE115746(mouse), and PHS001790(human). For the cross platform datasets, we employed PBMC(10x v2), PBMC(Smartseq2), PBMC(InDrop), PBMC(10x v3), PBMC(CEL-Seq2), PBMC(Seqwell), GSE84133(InDrop), GSE85241(CEL-Seq2), GSE81608(Smartseq2), and E-MTAB-5061(Smartseq2).

Author action: In our experiments, we utilized three distinct types of datasets and provided detailed descriptions of these datasets in the manuscript.

Reviewer#3, Concern#2: “The authors intend to present their method as more effective in single-cell identification. However, they do not provide the advantages/limitations of the more commonly used methods. For readers unfamiliar with the methodology, this information is relevant to understanding the manuscript results. Also, the methods used in comparison are not presented in the Materials and Methods section.”

Author response: Thanks. This is needed.

Author action: In the third and fourth paragraphs of the "Introduction" section, we added more commonly used methods, including the advantages and limitations of Seurat v3, Conos, scmap, CHETAH, and scGCN. Additionally, we provided an introduction to the comparative methods in the "Materials and Methods" section.

Reviewer#3, Concern#3: The second paragraph of the Introduction section presents many inherent limitations of the RNA-seq and/or single-cell technologies, not exclusive from the analysis. Since the authors propose to present an alternative to the analysis, not for the technology itself, I believe the analysis limitations should be better detailed instead of the technology ones.”.

Author response: Thank you for your feedback. In the second paragraph of the "Introduction" , we discussed the inherent limitations of the RNA-seq and/or single-cell technologies. Additionally, in the third and fourth paragraphs of the "Introduction", we analyzed the limitations of commonly used data analysis methods and then proposed our method to overcome the limitations of existing methods.

Author action: In the third and fourth paragraphs of the "Introduction," we have provided a detailed analysis of the limitations of existing analysis methods and have proposed our method to overcome these limitations.

Reviewer#3, Concern#4: “What were the criteria for the dataset selection? GSE84133, as I mentioned, has only 4 human samples and only 2 mouse samples. GSE115746 is from the cortex, whilst GSE84133 is from pancreatic islets. What is the biological context behind tissue selection?”

Author response: With the increasing availability of single-cell datasets, there is a pressing need to utilize existing and newly generated data in a reliable and reproducible manner, learn from well-defined and annotated established single-cell datasets, and transfer annotations to newly generated datasets for cell-level annotation. However, existing and newly generated datasets are often collected from different tissues and species, under various experimental conditions, through different platforms, and across different omics types. To meet the needs of practical applications, we need to perform experiments on cells from different tissues and species. Therefore, we selected cortical cells and pancreatic beta cells from humans and mice for our experiments.

Author action: In the "Data Collection" section, we added the significance and criteria for dataset selection.

Reviewer 4 Report

Comments and Suggestions for Authors

The study related to single-cell RNA sequencing (scRNA-seq), focusing on cell type identification from scRNA-seq data, is quite relevant. However, it requires some adjustments, and the main points are:

Introduction

Broader Contextualization: Emphasize the importance and impact of single-cell RNA sequencing (scRNA-seq) in biomedical research and personalized medicine. This might include a discussion on how scRNA-seq has transformed our understanding of cellular heterogeneity and its application in areas such as oncology, neuroscience, and the development of targeted therapies. This broader contextualization will help readers better understand the relevance and applicability of the study.

Comparison with Previous Works: Elaborate a more detailed comparison between the proposed method (scMGCN) and other existing approaches at the beginning of the introduction. This could include a discussion about the limitations of current methods, such as the difficulty in dealing with cell heterogeneity and batch effects in scRNA-seq data. Highlighting these limitations and how scMGCN addresses or overcomes them not only establishes the necessity of the research but also positions the study within the existing field, offering a clear justification for the development of the new method.

Materials and Methods

Deeper Detailing of Graph Construction Methods: Expand the explanation of the different graph construction methods used to generate input data for the scMGCN model. This would include a more detailed description of how each method contributes to capturing different aspects of scRNA-seq data and how they complement each other in the context of the multi-view model. Providing more details on the underlying principles of each method and the rationale for choosing these specific methods can help readers better understand how scMGCN processes and interprets the data.

Clearer Explanation of Model Implementation and Parameterization: Offer more detailed information about the implementation of the scMGCN model, including technical details such as hyperparameter configuration, training process, and validation criteria. This could involve describing the choices made for the network architecture, such as the number of layers and units in each layer, the activation functions used, and the optimization and regularization methods employed. It might also be useful to explain how the hyperparameters were adjusted and validated, and whether any specific cross-validation technique or test data set was used to assess the model's performance.

Discussion

Detailed Analysis of Limitations: Although the scMGCN model has demonstrated effectiveness, it is crucial to discuss its limitations in more detail. This includes limitations related to the quality of input data, the model's sensitivity to data variations, and challenges in interpreting the results. Exploring these limitations helps to contextualize the results and guide future research to overcome them.

Comparison with Similar Studies: Include a more in-depth comparison with other studies that use similar methods or address similar problems. This can help position scMGCN within the broader field of research in systems biology and machine learning. Discussing differences in results, methodologies, and applications can provide valuable insights into the advantages and disadvantages of scMGCN compared to other approaches.

Implications and Practical Applications: Expand the discussion on the practical implications of the model. How can the results be applied in real-world contexts, such as in biomedical research, treatment development, or understanding complex diseases? Illustrating with specific scenarios where scMGCN could have a significant impact can make the discussion more relevant and applicable.

Suggestions for Future Research: Offer clear recommendations for future research, including how the scMGCN model can be improved, adapted, or extended to other types of data or biological problems. This might involve exploring new network architectures, integrating with other data modalities, or testing in different biological contexts.

Reflection on Challenges and Trends in the Field: Include a discussion about the general challenges faced in the field of scRNA-seq analysis and how the scMGCN model fits into current research trends. This can help highlight the relevance and opportunity of the study in the broader context of data science and systems biology.

Comments on the Quality of English Language

The English quality of the provided translation is good. It accurately conveys the intended meaning of the original text in a clear and understandable manner. The sentence structure, grammar, and vocabulary are appropriate for academic or professional contexts. If you have specific sentences or sections you'd like to evaluate or improve further, feel free to share them!

Author Response

Thank you very much for your valuable and meaningful comments. These comments have provided great help for our revision process. We appreciate your patience and continuous help.

Reviewer#4, Concern#1: “Broader Contextualization: Emphasize the importance and impact of single-cell RNA sequencing (scRNA-seq) in biomedical research and personalized medicine. This might include a discussion on how scRNA-seq has transformed our understanding of cellular heterogeneity and its application in areas such as oncology, neuroscience, and the development of targeted therapies. This broader contextualization will help readers better understand the relevance and applicability of the study.”

Author response: I'm sorry, we did not provide a more detailed background on single-cell RNA sequencing technology in the "Introduction" section. We have now described the importance and impact of single-cell RNA sequencing in biomedical research and personalized medicine in the first paragraph of the "Introduction".

Author action: In the first paragraph of the "Introduction" section, we described the impact of single-cell RNA sequencing in biomedical research, particularly its applications in oncology.

Reviewer#4, Concern#2: “Comparison with Previous Works: Elaborate a more detailed comparison between the proposed method (scMGCN) and other existing approaches at the beginning of the introduction. This could include a discussion about the limitations of current methods, such as the difficulty in dealing with cell heterogeneity and batch effects in scRNA-seq data. Highlighting these limitations and how scMGCN addresses or overcomes them not only establishes the necessity of the research but also positions the study within the existing field, offering a clear justification for the development of the new method.”

Author response: I'm sorry, we did not provide a detailed account of the limitations of previous work. We have now addressed this in the third and fourth paragraphs of the "Introduction" section.

Author action: In the third and fourth paragraphs of the "Introduction" section, we added relevant information on the limitations of current related methods.

Reviewer#4, Concern#3: “Deeper Detailing of Graph Construction Methods: Expand the explanation of the different graph construction methods used to generate input data for the scMGCN model. This would include a more detailed description of how each method contributes to capturing different aspects of scRNA-seq data and how they complement each other in the context of the multi-view model. Providing more details on the underlying principles of each method and the rationale for choosing these specific methods can help readers better understand how scMGCN processes and interprets the data.”

Author response: We apologize for not providing in-depth descriptions of the details of graph construction methods. We have now added relevant supplementary materials with detailed descriptions of the graph construction process.

Author action: We added detailed information on the graph construction method to the Supplementary Materials of the article.

Reviewer#4, Concern#4: “Clearer Explanation of Model Implementation and Parameterization: Offer more detailed information about the implementation of the scMGCN model, including technical details such as hyperparameter configuration, training process, and validation criteria. This could involve describing the choices made for the network architecture, such as the number of layers and units in each layer, the activation functions used, and the optimization and regularization methods employed. It might also be useful to explain how the hyperparameters were adjusted and validated, and whether any specific cross-validation technique or test data set was used to assess the model's performance.”

Author response: We apologize for not providing a clearer explanation of the model implementation and parameterization. We have now described the specific implementation process of the model, parameter configuration, and the cross-validation technique used in the Supplementary Materials.

Author action: We have added detailed descriptions of the specific implementation details of the model, parameter configurations, and the cross-validation techniques used in the Supplementary Materials.

Reviewer#4, Concern#5: “Detailed Analysis of Limitations: Although the scMGCN model has demonstrated effectiveness, it is crucial to discuss its limitations in more detail. This includes limitations related to the quality of input data, the model's sensitivity to data variations, and challenges in interpreting the results. Exploring these limitations helps to contextualize the results and guide future research to overcome them.”

Author response: We apologize for not discussing the limitations of the scMGCN model in detail in the discussion. We have now added a section on the limitations of the scMGCN model to the "Discussion" section.

Author action: We have added a description of the current limitations of scMGCN in the "Discussion" section.

Reviewer#4, Concern#6: “Comparison with Similar Studies: Include a more in-depth comparison with other studies that use similar methods or address similar problems. This can help position scMGCN within the broader field of research in systems biology and machine learning. Discussing differences in results, methodologies, and applications can provide valuable insights into the advantages and disadvantages of scMGCN compared to other approaches.”

Author response: We apologize for not including a comparison between similar methods and the scMGCN method in the "Discussion" section. We have now compared the scMGCN method with other current mainstream methods in the "Discussion" section.

Author action: We added comparison between similar methods and scMGCN in the "Dissusion" section, highlighting the limitations of the similar methods and how scMGCN can overcome them.

Reviewer#4, Concern#7: “Suggestions for Future Research: Offer clear recommendations for future research, including how the scMGCN model can be improved, adapted, or extended to other types of data or biological problems. This might involve exploring new network architectures, integrating with other data modalities, or testing in different biological contexts.”

Author response: We apologize for not discussing future research suggestions in the "Discussion" section. We have now discussed the current issues with scMGCN and provided relevant improvement suggestions and directions in the "Discussion" section.

Author action: In the "Discussion" section, we have discussed the current issues with scMGCN and provided suggestions on how to improve and extend it to address these issues.

Reviewer#4, Concern#8: “Reflection on Challenges and Trends in the Field: Include a discussion about the general challenges faced in the field of scRNA-seq analysis and how the scMGCN model fits into current research trends. This can help highlight the relevance and opportunity of the study in the broader context of data science and systems biology.”

Author response: We apologize for not discussing some of the challenges in the field of scRNA-seq analysis and how scMGCN can address these challenges in the "Discussion" section. We have now added a discussion on the existing challenges in scRNA-seq analysis and the strategies that scMGCN can adopt to address these challenges in the "Discussion" section.

Author action: In the "Discussion" section, we have added a description of the challenges in the field of scRNA-seq analysis and how scMGCN can adapt to current research trends.

Round 2

Reviewer 2 Report

Comments and Suggestions for Authors

The authors have modified the text according to my suggestions.

However, the most relevant benchmarking comparison is still missing, as the authors did not report data for scGCN in Table 5. Since this is the major competitor (and scMGCN offers actually only a marginal increment (in most cases ~1% increase in accuracy) over scGCN, the authors should compare the memory requirements and GPU time for the two softwares.

The authors should also clearly highlight in the discussion (unless the benchmarking reveals that scGCN requires substantially more computational resources compared to scMGCN) that the increase in performance (1% more accuracy) over existing software with similar hardware requirements is in fact small.

minor points: I could not open the supplementary data archive, it seems to be corrupted.

Author Response

Thank you very much for your valuable and meaningful comments. These comments have provided great help for our revision process. We appreciate your patience and continuous help.

Reviewer#2,Concern#2: However, the most relevant benchmarking comparison is still missing, as the authors did not report data for scGCN in Table 5. Since this is the major competitor (and scMGCN offers actually only a marginal increment (in most cases ~1% increase in accuracy) over scGCN, the authors should compare the memory requirements and GPU time for the two softwares.

Author response: Because the comparison results of the running time between scGCN and Seurat in the scGCN paper [1] showed that the running time of scGCN is higher than that of Seurat, we did not evaluate the running time of scGCN in our last revision. We are very sorry for this. In Table 5, we have added relevant data on the storage and training time consumption of scGCN.

Author action: We have already added the storage and time consumption of scGCN to Table 5.

Reviewer#2,Concern#2: “The authors should also clearly highlight in the discussion (unless the benchmarking reveals that scGCN requires substantially more computational resources compared to scMGCN) that the increase in performance (1% more accuracy) over existing software with similar hardware requirements is in fact small.”

Author response: Thanks. This is needed.

Author action: In the first paragraph of the Discussion section, we emphasized that in some cases, the improvement of scMGCN compared with other methods is not obvious, but in other cases, the improvement of scMGCN is more significant than other methods.

Reviewer#2,Concern#3: “minor points: I could not open the supplementary data archive, it seems to be corrupted.”

Author response: We apologize for this, and the relevant documents have been re-uploaded.

Author action: We have re-uploaded the relevant files.

Reference

[1] Song, Q.; Su, J.; Zhang, W. scGCN is a graph convolutional networks algorithm for knowledge transfer in single cell omics. Nature communications 2021, 12, 3826.

Reviewer 3 Report

Comments and Suggestions for Authors

The authors performed alterations in the text that improved the understanding of the main results. However, I still believe they should not use datasets with such a small sample size to validate an analysis method. Even though other datasets are used, and the number of cells is high, sample collection and processing might have biases that cannot be corrected, considering there are only 2 mouse entries.

I also suggest a column with the sample size (not only the number of cells) be added to Table 6.

Although they have added more information regarding the study selection, the authors have still not answered why they have decided to use these datasets and not others available in GEO. I would like to read about the biological sense behind this selection, not only platform and methodological aspects. What is the importance of using these tissues and not others? Are there biological differences (sex, age) that could interfere with the analysis?

Author Response

Thank you very much for your valuable and meaningful comments. These comments have provided great help for our revision process. We appreciate your patience and continuous help.

Reviewer#3,Concern#1: “The authors performed alterations in the text that improved the understanding of the main results. However, I still believe they should not use datasets with such a small sample size to validate an analysis method. Even though other datasets are used, and the number of cells is high, sample collection and processing might have biases that cannot be corrected, considering there are only 2 mouse entries.

Although they have added more information regarding the study selection, the authors have still not answered why they have decided to use these datasets and not others available in GEO. I would like to read about the biological sense behind this selection, not only platform and methodological aspects. What is the importance of using these tissues and not others? Are there biological differences (sex, age) that could interfere with the analysis?”

Author response: Thank you for your suggestion. Our main contribution is proposing an algorithm for cell identification in single-cell transcriptome data. To validate the algorithm's performance, we used benchmark datasets previously used by some of the classic methods [1-5]. These methods validated the performance of cell type recognition on single datasets such as GSE115746, PHS001790, GSE118389, GSE72056, and GSE98638, cross-species datasets such as GSE84133 (mouse), GSE84133 (human), GSE115746 (mouse), and PHS001790 (human), and cross-platform datasets such as PBMC (10x v2), PBMC (Smartseq2), PBMC (InDrop), PBMC (10x v3), PBMC (CEL-Seq2), PBMC (Seqwell), GSE84133 (InDrop), GSE85241 (CEL-Seq2), GSE81608 (Smartseq2), and E-MTAB-5061 (Smartseq2). So, we also compared the performance of our algorithm on datasets with small samples as those used in previous studies, using multiple types of datasets for validation performance.

Author action: In the final sentence of the third paragraph within the section on Data Collection, we have explained the reasons behind our choice of these datasets.

Reviewer#3,Concern#2: “I also suggest a column with the sample size (not only the number of cells) be added to Table 6.”

Author response: Thanks. This is needed.

Author action: We have added a column for cell type and a column with sample size in Table 6.

[1] Stuart, T.; Butler, A.; Hoffman, P.; Hafemeister, C.; Papalexi, E.; Mauck III, W.M.; Hao, Y.; Stoeckius, M.; Smibert, P.; Satija, R. Comprehensive integration of single-cell data. Cell 2019, 177, 1888–1902.

[2] Kiselev, V.Y.; Yiu, A.; Hemberg, M. scmap: projection of single-cell RNA-seq data across data sets. Nature methods 2018, 15, 359–362.

[3] Song, Q.; Su, J.; Zhang, W. scGCN is a graph convolutional networks algorithm for knowledge transfer in single cell omics. Nature communications 2021, 12, 3826.

[4] Abdelaal, T.; Michielsen, L.; Cats, D.; Hoogduin, D.; Mei, H.; Reinders, M.J.T.; Mahfouz, A. A comparison of automatic cell identification methods for single-cell RNA sequencing data. Genome Biology 2019, 20.

[5] Butler, A.; Hoffman, P.J.; Smibert, P.; Papalexi, E.; Satija, R. Integrating single-cell transcriptomic data across different conditions, technologies, and species. Nature Biotechnology 2018, 36, 411–420.

Reviewer 4 Report

Comments and Suggestions for Authors

The manuscript has shown improvement in the necessary areas, effectively highlighting the limitations of the current model and suggesting future research directions, which is indicative of scientific rigor and integrity. Additionally, the code and related datasets are publicly available, facilitating verification and further research by other scientists. Therefore, I believe the manuscript is ready for publication.

Comments on the Quality of English Language

The language is clear, technical, and appropriate for a scientific article. It effectively communicates the purpose, methods, and significance of the research. 

Author Response

Thank you very much for your valuable and meaningful comments. These comments have provided great help for our revision process. We appreciate your patience and continuous help.

Reviewer#4,Concern#1: “The manuscript has shown improvement in the necessary areas, effectively highlighting the limitations of the current model and suggesting future research directions, which is indicative of scientific rigor and integrity. Additionally, the code and related datasets are publicly available, facilitating verification and further research by other scientists. Therefore, I believe the manuscript is ready for publication.”

Author response: I appreciate your acknowledgment of our efforts.

Reviewer#4,Concern#2: “The language is clear, technical, and appropriate for a scientific article. It effectively communicates the purpose, methods, and significance of the research.”

Author response: I appreciate your acknowledgment of our efforts.

Round 3

Reviewer 2 Report

Comments and Suggestions for Authors

The authors have addressed all of my concerns.

Author Response

Thank you very much for your valuable and meaningful comments. These comments have provided great help for our revision process. We appreciate your patience and continuous help.
Reviewer#1,Concern#1:“The authors have addressed all of my concerns.”
Author response: Thank you for your affirmation and assistance in our work.
Author action: We have further revised and refined the English sentences in the manuscript.

Reviewer 3 Report

Comments and Suggestions for Authors

Although the authors have justified their strategy, I still believe the sample size should be increased. The dataset selection must be justified in the biological sense, not the methodological one (sc vs. bulk RNA-Seq).

Author Response

Thank you very much for your valuable and meaningful comments. These comments have provided great help for our revision process. We appreciate your patience and continuous help.

Reviewer#1,Concern#1:“Although the authors have justified their strategy, I still believe the sample size should be increased. The dataset selection must be justified in the biological sense, not the methodological one (sc vs. bulk RNA-Seq).”

Author response: We have removed the dataset GSE84133, which contained a relatively small sample size, and added new datasets with larger sample sizes, including GSE85241, GSE109774, SRP073767, GSE120221, and GSE107727.

Author action: In our article, we have removed the dataset GSE84133, which contained a relatively small sample size. Instead, we have selected new datasets, namely GSE85241, GSE109774, SRP073767, GSE120221, and GSE107727, and conducted relevant experiments using our proposed model. We have also provided an explanation of their effects in the article.